# QuaDMix: Quality-Diversity Balanced Data Selection for Efficient LLM Pretraining

## Abstract

Quality and diversity are two critical metrics for the training data of large language models (LLMs), positively impacting performance. Existing studies often optimize these metrics separately, typically by first applying quality filtering and then adjusting data proportions. However, these approaches overlook the inherent trade-off between quality and diversity, necessitating their joint consideration. Given a fixed training quota, it's essential to evaluate both the quality of each data point and its complementary effect on the overall dataset. In this paper, we introduce a unified data selection framework called QuaDMix, which automatically optimizes the data distribution for LLM pretraining while balancing both quality and diversity. Specifically, we first propose multiple criteria to measure data quality and employ domain classification to distinguish data points, thereby measuring overall diversity. QuaDMix then employs a unified parameterized data sampling function that determines the sampling probability of each data point based on these quality and diversity related labels. To accelerate the search for the optimal parameters involved in the QuaDMix framework, we conduct simulated experiments on smaller models and use LightGBM for parameters searching, inspired by the RegMix method. Our experiments across diverse models and datasets demonstrate that QuaDMix achieves an average performance improvement of 7.2% across multiple benchmarks. These results outperform the independent strategies for quality and diversity, highlighting the necessity and the framework's ability to balance data quality and diversity.

## 1 Introduction

The efficiency and preference of pretraining large language models are significantly influenced by the characteristics of the training corpus (Brown et al., 2020; Chowdhery et al., 2023; Longpre et al., 2024). There is evidence from existing research suggesting that the model performance can be improved through the curation of high-quality data (Wettig et al., 2024; Xie et al., 2023b; Sachdeva et al., 2024), the application of data deduplication and diversification strategies (Abbas et al., 2023; Tirumala et al., 2023), and the careful balancing of data distribution across various domains and topics (Liu et al., 2024; Xie et al., 2023a). Nevertheless, identifying optimal configuration of combining those factors remains an open challenge, due to complex interplay between data quality and diversity, which has yet to be fully understood.

There remains two major challenges to identify the optimal data selection strategy. Firstly, the definition of quality and diversity is ambiguous. Previous research has proposed various definitions of quality criteria, including factors such as regular expression (Penedo et al., 2023; Wenzek et al., 2020), educational value (Penedo et al., 2024), similarity to instruction tuning data (Li et al., 2024), etc, each emphasizing only a specific aspect of the data. On the other hand, approaches like (Liu et al., 2024; Abbas et al., 2023) optimize the data mixtures for more effective training, indicating that a better diversity is not necessarily uniform distribution. Secondly, there exists interplay between data quality and diversity. The choice of quality criteria affects the distribution of selected data as illustrated in Figure 1b, due to inherent biases in different criteria. Meanwhile, changing of data mixtures influences the data quality, as the quality level differs across different domains. Also, since the high quality data is limited, the trade-off between better quality or diversity is inevitable, which is not feasible by optimizing only for data quality or diversity. How to jointly optimize the data distribution together with the selection of quality criteria remains another unsolved issue.

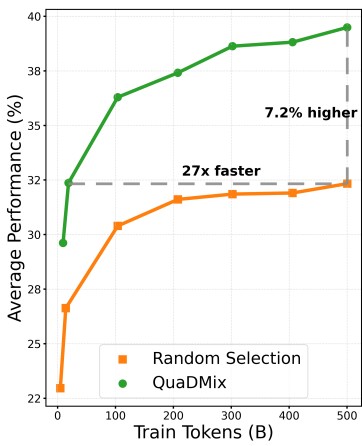

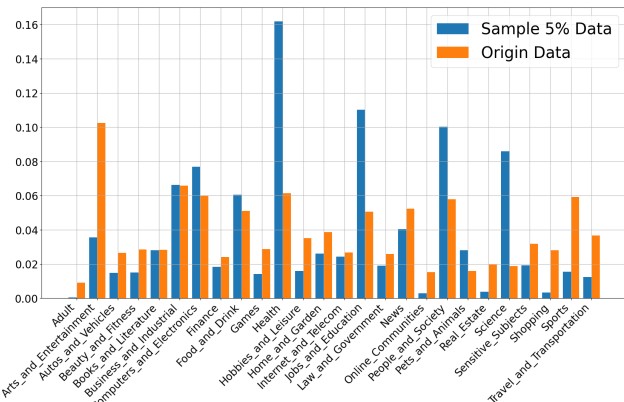

(a) Comparison between QuaDMix and random selection on average performance of 9 dnowstream tasks of 530M model trained from scratch.

(b) The distribution change of data selected with Fineweb-edu Classifier. With the top5% documents selected, the ratio of certain domains including Health, Jobs and Education, increases for a large margin compared with original data.

To address these challenges, we propose a unified data selection framework, QuaDMix, which simultaneously manages data quality and diversity. Firstly, we apply several quality scorers and domain classification on each document in the training corpus, to measure the data quality and diversity. Then a parameterized function is designed to determine the sampling frequency for each document based on those quality and domain labels. Specifically, an aggregated quality score is first computed by weighted averaging the quality scores, where the weights are controlled by adjustable parameters. Then a parameterized sampling function takes the aggregated quality score as input and calculate the sampling frequency, where data with higher quality is assigned with more frequency and the parameters affect how the frequency decreases as the quality diminishes. Here we take the assumption that training samples with higher quality worth sampled for more times. We assign independent parameters for data across different domains to control the diversity via parameters. To find the optimal parameters among the numerous parameter space, we employ a two-step approach inspired by (Liu et al., 2024). First, we train a set of small models on datasets sampled using QuaDMix with various parameter configurations, as an approximation for the performance of larger models. Next, we train a regression model to fit the performance results from this limited set of small models. This regression model is then used to predict the performance for unseen parameter configurations, providing an efficient way to explore the parameter space without exhaustive large-scale training.

To validate the effectiveness of QuaDMix, we train 3000 models with 1M parameters for 1B tokens, each using data sampled from RefinedWeb (Penedo et al., 2023) with various QuaDMix parameters. The optimal parameter configuration is then determined by searching the input space of a trained LightGBM regressor(Ke et al., 2017). We then evaluate different pretraining data selection methods on models with 530M parameters. The optimal configuration identified by QuaDMix achieves superior performance on an aggregated benchmark. Our results also reveal the following insights: (1) Different quality criteria exhibit trade-offs across downstream tasks, but appropriately merging these criteria yields consistent improvements across tasks by leveraging complementary information. (2) The optimal data mixture varies under different quality criteria, indicating the importance of jointly optimizing both the quality and diversity. (3) The target of regression model can guide the preference for specific downstream tasks, enabling task-focused data selection.

## 2 RELATED WORK

### 2.1 PRETRAINING DATA SELECTION

Data quality, diversity, and coverage are critical factors for ensuring the efficiency and generalizability of large language models (Cheng et al., 2024; Touvron et al., 2023; Chowdhery et al., 2023).

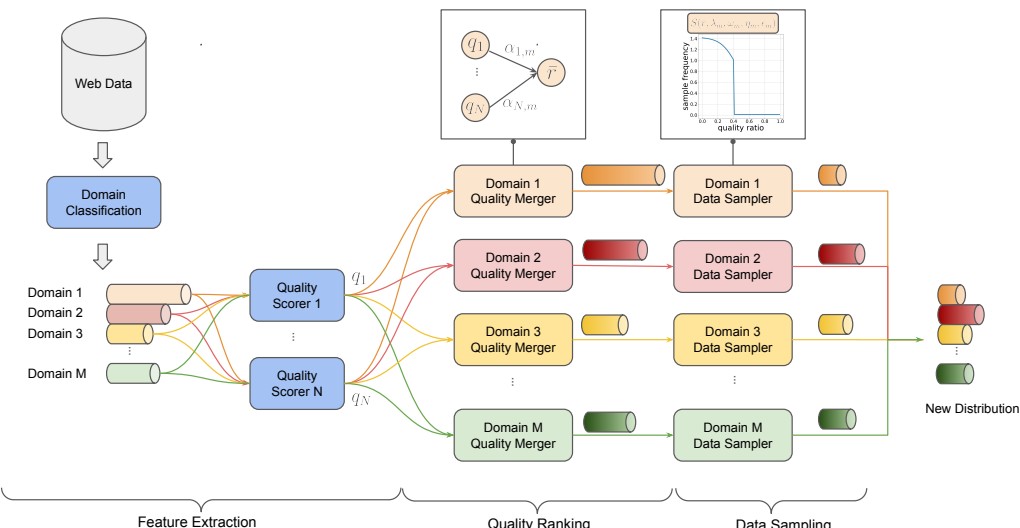

Figure 2: The overall design of QuaDMix. First we extract the data features using classifier and quality scores (QS). Then we calculate quality rank for each domain with the merging parameters. Finally the sampling functions controlled by sampling parameters are applied to generate the final output data.

To improve data quality, rule-based filtering techniques are commonly employed (Laurençon et al., 2022; Weber et al., 2024; Penedo et al., 2023; Raffel et al., 2020). These methods use handcrafted heuristics, such as removing terminal marks, detecting sentence repetitions, and enforcing length constraints, to exclude low-quality data. While these rules effectively filter out noisy data from the training corpus, they fail to capture semantic-level information, which is crucial for more refined data selection. Alternative approaches aim to address this limitation. For instance, (Wenzek et al., 2020; Marion et al., 2023; Thrush et al., 2024) use model perplexity as a measure of data quality, while (Lin et al., 2025) apply token-level selection by re-weighting the loss across tokens. (Xie et al., 2023b) utilize n-gram features to quantify data importance and sample accordingly. Discriminator-based methods (Brown et al., 2020; Du et al., 2022; Gao et al., 2020; Soldaini et al., 2024; Li et al., 2024) select data by comparing it to predefined high-quality datasets, such as Wikipedia or instruction-tuning datasets. However, how much these predefined datasets represent for high-quality relies on empirical judgement. More recently, approaches like (Gunasekar et al., 2023; Sachdeva et al., 2024; Wettig et al., 2024; Penedo et al., 2024) leverage large language models (e.g., GPT-4) to evaluate and filter data based on designed prompts that emphasize various dimensions of value, offering a more nuanced way to define and curate high-quality data.

To optimize data distribution, various methods leverage clustering and representativeness to achieve deduplication and diversification. For example, (Abbas et al., 2023; Shao et al., 2024; Tirumala et al., 2023) employ data clustering techniques to identify and select representative data points, ensuring both diversity and efficiency in the training corpus. Other approaches estimate optimal data mixtures through iterative modeling. (Xie et al., 2023a) first train a small reference model and subsequently optimize the worst-case loss across domains by training a proxy model to identify the optimal data mixture. Similarly, (Bai et al., 2024; Yu et al., 2024; Fan et al., 2024; Gu et al., 2024) calculate influence scores by tracking first-order gradients on an evaluation set, thereby identifying the most valuable data for training. Additionally, (Liu et al., 2024; Ye et al., 2024) simulate the performance of different data mixtures by training a series of proxy models, enabling the prediction of large-model performance with low compute cost.

## 2.2 SCALING LAWS

Neural Scaling Laws have been shown to effectively predict performance across varying training budgets, model sizes, and dataset scales in LLM pretraining (Kaplan et al., 2020; Rae et al., 2022). However, in practical scenarios where dataset size is limited, or data mixtures vary, scaling laws

exhibit significant variations (Hoffmann et al., 2022). Several studies have extended scaling laws to account for these complexities. (Muennighoff et al., 2023; Hernandez et al., 2022) explore the impact of data repetition levels on scaling behaviors, while (Ge et al., 2024) investigate scaling dynamics under different domain proportions and dataset sizes. To optimize data compositions, (Liu et al., 2024) propose a regression model for predicting optimal mixtures, and (Kang et al., 2024) further analyze optimal compositions across varying scales. Additionally, (Que et al., 2024) focus on identifying the best data mixtures for the continued pretraining stage, providing insights into refining pretraining strategies under diverse constraints.

## 3 METHODOLOGY

Our approach can be illustrated in 4 parts: 1) We propose the QuaDMix framework, which utilizes a unified parameterized function to govern the data sampling process. 2) We conduct small-scale experiments to explore how different parameter settings within QuaDMix affect the performance of LLM. 3) We train a regression model to capture these effects, using it to identify the optimal parameters. 4) With the optimal parameter settings, we sample large-scale data and train a large language model.

### 3.1 DESIGN OF QUADMIX

We design QuaDMix as a sampling algorithm that simultaneously accounts for data quality and diversity, as shown in Figure 2. Given a pretraining dataset $X$, we define a sampling function $S(x, \boldsymbol{q}_x, d_x; \boldsymbol{\theta})$, which determines the expected sampling times of each data point $x$ based on its data feature $\boldsymbol{q}_x$ and $d_x$. Here $\boldsymbol{q_x}$ represents the quality score vector, which includes multiple quality criteria, and $d_x$ denotes the domain to which $x$ belongs. $\boldsymbol{\theta} = (\alpha, \beta)$ are the merging and sampleing parameters to be optimized. The output of this function is fractional value, e.g. $a.b$, meaning the document will be sampled for $a$ times plus another random sampling with probability $b$.

**Feature Extraction** To measure a sample's contribution to diversity and its quality, we propose using domain classification and $N$ quality scorers to label the pretraining data. Specifically, we use a domain classifier to divide the dataset into $M$ domains, where $x$ will be assigned a domain label $d_x$. Then we use $N$ quality scorers to compute the quality vector $\boldsymbol{q_x} = (q_{1,x}, ..., q_{N,x})$, and for each $q_{n,x}$, a smaller value indicates a better quality on that dimension. For the sake of simplicity, we omit $x$ in the subscript in the following discussion.

**Quality Ranking** We first define a merging function that integrates the scores from various quality filters, aiming to incorporate complementary information provided by different criteria. Assuming there are $N$ criteria, for any individual example $\boldsymbol{x}$ belonging to domain $m$, the merged quality score is calculated by

$$\bar{q} = \sum_{n=1}^{N} \sigma(q_n)\alpha_{n,m},$$

(1)

where $\boldsymbol{\alpha}_m$ are the merging parameters for domain $m$. We utilize separate merging parameters to balance the quality criteria across different domains, as the criteria exhibit varying preferences depending on the domain. $\sigma$ is a normalization function to align the scales of quality criteria.

We then sort the data based on the merged quality score. The sorting is operated separately in each domain. The merged quality rank $\bar{r}$ calculated by computing the percentile of the data within that domain. That is

$$\bar{r} = \frac{|\{x|d_x = m, \bar{q}_x <= \bar{q}\}|}{|\{x|d_x = m\}|}.$$

(2)

Here we calculate the size of the set by adding up the number of tokens for all sample within the set. For a given example in domain $m$ with $\bar{r} = 0.05$, this means that 95% of the tokens in that domain have a worse quality compared to this example. (Note that we use smaller quality scores to represent higher quality.)

**Quality Sampling** Next, we define the sampling function. We take the assumption that higher-quality data should be sampled more frequently in the final dataset. This assumption is supported by

evidence (Penedo et al., 2024), which demonstrates that applying a higher quality threshold improves downstream performance. For any example in domain $m$ with merged quality rank $\bar{r}$, the value of the sampling function is determined by

$$S(\bar{r}) = \begin{cases} \left(\frac{2}{1+e^{-\lambda_m(\omega_m-\bar{r})}}\right)^{\eta_m} + \epsilon_m, & \bar{r} <= \omega_m \\ \epsilon_m, & \bar{r} > \omega_m \end{cases} \tag{3}$$

We denote $\boldsymbol{\beta}_m = (\lambda_m, \omega_m, \eta_m, \epsilon_m)$ as the sampling parameters for domain $m$. We use a format of sigmoid to ensure the sampling value is monotonically decreasing as the quality rank goes up (worse quality) and $\lambda_m$ is used to adjust how fast it decreases. $\omega_m$ controls the quality percentile threshold, determining the minimum quality level we aim to retain. $\eta_m$ is a scaling parameter that adjusts the sampling values, while $\epsilon_m$ introduces randomness to incorporate data from all quality ranges. By applying different sampling parameters across domains, we achieve flexible control over domain proportions.

In summary, by integrating (1),(2), and (3), we define the sampling function for individual domain $m$, with the parameters structured as $\boldsymbol{\theta}_m = (\boldsymbol{\alpha}_m, \boldsymbol{\beta}_m)$. The total number of parameters is $(N + 4) \times M$, where $N$ represents the number of used quality criteria and $M$ denotes the total number of distinct domains.

### 3.2 Proxy Model Experiments

We first sample a set of values for each parameter defined above, subsequently generating corresponding datasets using the QuaDMix sampling function. Following this, a series of small proxy models are trained on each dataset and evaluated on the validation set to compute the validation loss.
**Parameter Sampling** The parameter space requires careful design to encompass valuable regions, while avoiding extreme conditions. We sample from the parameter space as following:

---

**Algorithm 1** Parameter Sampling for QuaDMix

**Ensure:** $\boldsymbol{\theta}$
**Require:** $N, M$
  Sample $(a_1, ..., a_N) \sim U(0, 1)$
  $\tilde{a}_n = \frac{a_n}{\sum_i a_i}$
  **for** $m = 1$ **to** $M$ **do**
    Sample $(b_{1,m}, ..., b_{N,m}) \sim U(0, 1)$
    $\tilde{b}_{n,m} = \frac{\tilde{a}_n b_{n,m}}{\sum_i \tilde{a}_i b_{i,m}}$
    $\boldsymbol{\alpha}_m = (\tilde{b}_{n,m}), n = 1, ..., N$
    Sample $(\lambda_m, \omega_m, \eta_m, \epsilon_m) \sim U(0, 1)$
    $\tilde{\lambda}_m = 10^{3\lambda_m}, \ \tilde{\omega}_m = 0.1\omega_m$
    $\tilde{\eta}_m = \eta_m, \ \tilde{\epsilon}_m = \epsilon_m/1000$
    $\boldsymbol{\beta}_m = (\tilde{\lambda}_m, \tilde{\omega}_m, \tilde{\eta}_m, \tilde{\epsilon}_m)$
    $\boldsymbol{\theta}_m = (\boldsymbol{\alpha}_m, \boldsymbol{\beta}_m)$
  **end for**
  $\boldsymbol{\theta} = (\boldsymbol{\theta}_1, ..., \boldsymbol{\theta}_M)$

---

In the algorithm above, we introduce a global weight for each quality criteria, with the final weight computed by multiplying the global weight by the domain-specific weight. Without this global weight, the expected average weight across domains for each quality criterion would always be $1/N$, which fails to account for the scenario where one quality criterion may suppress another overall. For $\boldsymbol{\beta}_m$, we rescale them accordingly to ensure domain proportions and quality thresholds remain within a reasonable range. Using this process, we generate 3,000 sets of parameters $\boldsymbol{\theta}_i$ and then sample with QuaDMix from our training data, producing 3,000 proxy datasets, denoting as $D_i$.

**Proxy Model Training** Next we train the proxy models on each proxy datasets from scratch.

$$f_i^* = \arg\min_f L(f, D_i)$$

After training, we evaluate the proxy models by calculating the loss on the target evaluation datasets.

$$L_i = L(f_i^*, D_{eval})$$

### 3.3 PARAMETER OPTIMIZING

**Regression Model Fitting** The next step is to determine the correlation between the sampled QuaD-Mix parameters and model performance. We formulate this as a regression problem, as proposed in (Liu et al., 2024), with the goal of learning a function that predicts the target value based on the input features. Specifically, we optimize a regressor $R$ with

$$R^* = \arg \min_R \sum_i ||R(\boldsymbol{\theta}_i) - L_i||^2$$

We evaluate different types of regressors and select LightGBM (Ke et al., 2017), which ensembles multiple decision trees, to predict the target value.

**Optimal Parameter Estimation** Once the regressor is trained, we search the input space to find the optimal parameters that minimize the predicted loss. Rather than performing a random search across the entire space, we sample 100,000 data points using the algorithm outlined in Section 3.2 to mitigate the influence of outliers on the regressor. To further reduce the variance in the regression predictions, we sort the data points based on their predicted target values and calculate the average of the top 10 data points to determine the final output.

### 3.4 LARGE-SCALE MODEL EXPERIMENTS

We then use the optimal parameters to generate large-scale datasets for training large-scale models. In practice, since sorting the quality scores across the entire dataset is computationally expensive, we estimate the quality percentile by randomly selecting a subset of 10,000 documents. Within this subset, we calculate the mapping between the quality percentile and quality score, and then apply this mapping to the entire dataset.

## 4 EXPERIMENTS ON REGRESSION MODEL

### 4.1 EXPERIMENT SETUP

**Datasets** We conduct our experiment on RefinedWeb (Penedo et al., 2023). It is an English large-scale dataset for the pretraining of large language models and consists of over 570B(billion) tokens. For the small proxy datasets, we sample it from a subset of RefinedWeb, each containing 1B tokens.

**Feature Extraction** We generate the necessary data features including data quality and domain index with 3 individual quality filters, AskLLM (Sachdeva et al., 2024), Fineweb-Edu (Penedo et al., 2024), DCLM (Li et al., 2024) and 1 domain classifier (Jennings et al.), which classifies the data into 26 different domains with a Deberta V3 (He et al., 2023) architecture. The detail of the classifier is reported in Appendix A

**Training and evaluation** For the proxy models, we train them on the proxy datasets for 1B tokens. More information about proxy models are in Appendix B.

To construct the validation datasets, we sample from the instruction-formatted dataset OpenHermes 2.5 (Teknium, 2023). As demonstrated in (Li et al., 2024), this dataset is used to train a robust quality filter. To improve efficiency, we sampled 10k samples from it to form a validation subset, named openhermes-10k. Additionally, we test on the training data from the downstream tasks including HellaSwag, ARC-E, ARC-C, MMLU, and TriviaQA to demonstrate the model's ability to optimize for specific downstream tasks by altering the target evaluation datasets.

For the regression model, we split the data into 2800/200 for training and validation. We use Mean Absolute Error (MAE) as the evaluation metric, which calculates the average absolute differences between predicted and actual values.

## 4.2 RESULTS

We use LightGBM as the regression model, and train it with 2800 proxy model's results, and there is a strong correlation between the predicted loss and real loss on the 200 validation set, the pearson correlation is $95.45\%$. Indicating the LightGBM is able to predict the proxy model performance based on the inout parameters.

# 5 EXPERIMENTS ON LANGUAGE MODEL

In this section we compare different methods of data selection and mixture with QuaDMix by training language models from scratch and evaluating on various downstream tasks.

## 5.1 EXPERIMENT SETUP

**Training and evaluation** We train the language model with 530M parameters from scratch for 500B tokens. And we also conduct experiments on 1.2B model with 30B tokens and 7B model with 150B tokens to further validate the effectiveness of QuaDMix. We train modekl with transformer architecture (Vaswani et al., 2017), see the details in Appendix B and Appendix D.

Then we evaluate the model performance using lm-eval-harness (Gao et al., 2023). We choose 9 downstream tasks, including 3 commonsense reasoning tasks (PIQA (Bisk et al., 2019), HellaSwag (Zellers et al., 2019), OpenBookQA (Mihaylov et al., 2018)), 3 reading comprehension tasks (ARC-E/C (Clark et al., 2018), Triviaqa (Joshi et al., 2017)), 1 math problem solving task (SVAMP (Patel et al., 2021)) and 2 knowledge intensive tasks (MMLU (Hendrycks et al., 2021), NQ-open (Kwiatkowski et al., 2019; Lee et al., 2019)). For each benchmark, we used normalized accuracy as the evaluation metric. Some modifications on the testing logic are applied for numerical stability, the details are shown in Appendix C.

## 5.2 DATA SELECTION METHODS

We use the following methods as comparative experiments with our QuaDMix:

• **Random Selection**: Documents are randomly selected from the whole dataset.

• **Fineweb-edu Classifier**: Documents are scored with Fineweb-edu Classifier (Penedo et al., 2024) with top-k selection

• **AskLLM**: Documents are scored with the probability of generating "Yes" from a prompted large language model (Sachdeva et al., 2024). The top-k documents are selected.

• **DCLM**: Documents are scored with fasttext based classifier (Li et al., 2024) with top-k selection.

• **Criteria Mix**: Following (Wettig et al., 2024), the selected data from the above three filters are merged, with duplicated documents removed.

Table 1: QuaDMix outperforms the methods focusing only on data quality or data mixture. With benchmark training set as the target, the results further boost.

| Methods | Selected Token | Reading Comprehension | Commonsense Reasoning | Knowledge | Math | Average |
|---|---|---|---|---|---|---|
| Random Selection | 500B | 32.9 | 51.6 | 17.4 | 2.8 | 32.3 |
| DSIR | 72B | 34.9 | 49.2 | 17.5 | 6.9 | 32.7 |
| RegMix | 500B | 35.5 | 52.4 | 17.7 | 3.5 | 33.6 |
| Fineweb-edu | 30B | 41.4 | 55.5 | 20.1 | 6.0 | 37.4 |
| AskLLM | 30B | 38.9 | 54.2 | 19.0 | 2.3 | 35.5 |
| DCLM | 30B | 41.2 | 53.1 | 19.8 | 8.2 | 36.7 |
| Criteria Mix | 74B | 40.1 | 53.7 | 20.0 | 3.1 | 36.0 |
| QuaDMix-OH | 30B | 44.0 | 55.7 | 21.0 | 10.2 | 39.0 |
| QuaDMix-BMK | 30B | **44.8** | **55.7** | **21.3** | **11.5** | **39.5** |

• **DSIR**: Documents are sampled based on the importance calculated with the N-gram features (Xie et al., 2023b).

• **RegMix**: Following (Liu et al., 2024), we conduct 512 1M porxy experiments and randomly select data using the optimized data mixtures.

And we conduct QuaDMix experiments base on two different evaluation sets:

• **QuaDMix-OH**: Documents are sampled with the proposed QuaDMix, where Openhermes is used as the validation set for the proxy experiments

• **QuaDMix-BMK**: Documents are sampled with the proposed QuaDMix, where the training set of 5 downstream tasks (HellaSwag, ARC-E, ARC-C, MMLU, TriviaQA) are used as the validation set to generate the optimal QuaDMix parameters. See appendix G for the details.

## 5.3 RESULTS

The results are summerized in Table 1. We can see that QuaDMix outperforms the methods focusing only on data quality or data mixture on all the benchmarks, proving the necessity of jointly considering quality and diversity. It also shows that the proxy model experiments can well indicate the performance on large scale model. With loss of the benchmark training set as the target when training the regression model, the results further boost. This prove the ability of QuaDMix of optimizing for specific downstream tasks by choosing evaluation datasets in proxy experiments which are more related to the downstream tasks. The detailed experiment results are shown in Appendix E.

Figure 1a indicats QuaDMix is computaionally efficient, QuaDMix achieves the same average performance as the Random Selection baseline at 500B tokens with only about 18B tokens, which is a 27 times faster. Furthermore, QuaDMix surpasses the Random Selection baseline by 7.2% in average performance at 500B token.

**Analysis of optimal QuaDMix parameters** We show the optimal data mixtures and merging parameters of quality filters from QuaDMix-BMK in Figure 3. We see that the Health and Science domain are upsampled for large margin, while Sports and Computers downsampled, indicating that the downstream tasks we choose have preference for specific domains. The right figure shows that the DCLM quality filter contributes most to the merged quality score, while AskLLM only occupies a small weight among the three filters.

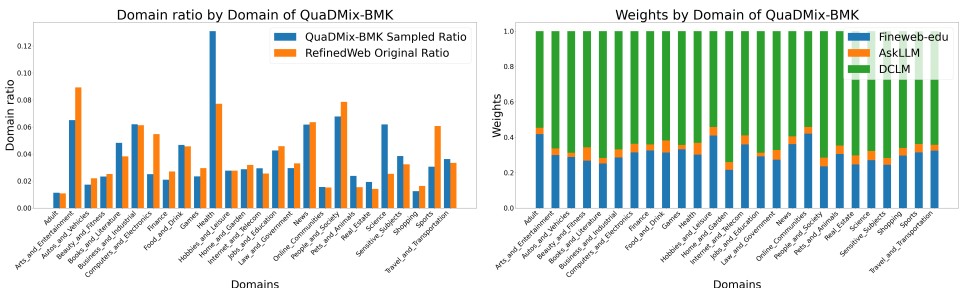

Figure 3: The visualization of optimal parameters from QuaDMix-BMK.

## 6 ABLATIONS

**Quality Merging Benefits Selection** To prove the necessity of quality score merging, we select different combinations of quality filters by manually setting the weight of certain filters to 0 when finding the optimal QuaDMix parameters. As shown in Table 2, merging with all three quality filters shows the best performance. Although using one quality filter can be optimal for one specific task, for example DCLM-only for MATH, the merging process reduces intrinsic bias within the quality filters and outperforms in general ability, which is essential for language model pretraining.

**More Tokens not always good** We also experiment with selecting more tokens by loosing the sampling parameter $\omega$ in QuaDMix. In that way we introduce more diversed tokens but lower

Table 2: QuaDMix-OH with different settings on quality filters (AskLLM (A), Fineweb-edu (F), DCLM (D)) and selected tokens.

| A | F | D | Selected Token | Reading Comprehension | Commonsense Reasoning | Knowledge | Math | Average |
|---|---|---|---|---|---|---|---|---|
| ✓ | | | 30B | 38.9 | 53.5 | 18.6 | 2.9 | 35.2 |
| | ✓ | | 30B | 41.4 | 55.5 | 20.1 | 6.0 | 37.4 |
| | | ✓ | 30B | 41.3 | 53.4 | 19.7 | **12.2** | 37.3 |
| ✓ | ✓ | | 30B | 41.9 | 55.6 | 20.0 | 5.1 | 37.5 |
| ✓ | | ✓ | 30B | 41.8 | 54.6 | 19.8 | 9.1 | 37.5 |
| | ✓ | ✓ | 30B | 43.5 | 55.6 | 20.8 | 9.6 | 38.7 |
| ✓ | ✓ | ✓ | 90B | 40.7 | 55.2 | 19.5 | 4.6 | 36.8 |
| ✓ | ✓ | ✓ | 180B | 37.8 | 53.9 | 18.9 | 2.8 | 35.1 |
| ✓ | ✓ | ✓ | 30B | **44.0** | **55.7** | **21.0** | 10.2 | **39.0** |

Table 3: QuaDMix-OH vs QuaDMix-BMK on 5 downstream tasks. The trend mostly agree with the prediction loss on proxy model except for HellaSwag.

| Method | HellaSwag | ARC-C | ARC-E | MMLU | TriviaQA |
|---|---|---|---|---|---|
| QuaDMix-OH | **56.5** | 39.2 | 71.1 | 34.1 | 21.6 |
| QuaDMix-BMK | 56.1 | **40.2** | **71.3** | **34.4** | **22.8** |

quality into the training. The results in Table 2 show that selecting 30B tokens, i.e. documents with top5% quality yields the best result, meaning that curing data quality contributes more than increasing the number of unique tokens within this range.

**Proxy Ability of Small Models** How well the prediction loss on proxy models forecasts the performance on large-scale models is the key factor of QuaDMix. To study this, we train 5 separate regression models, each using the loss on training set of one benchmark as the target. The results on the validation set are shown as blue points in Figure 4. We notice that HellaSwag has larger variance than others, which indicates there may be more influencing factors related with HellaSwag, making the loss on it harder to predict. Then we predict the loss for optimal parameters from QuaDMix-OH and QuaDMix-BMK using each regression model as shown in Figure 4. It is reasonable to see the loss of QuaDMix-BMK surpasses QuaDMix-OH on all tasks since QuaDMix-BMK utilizes benchmark training set as optimizing target. Finally we report the performance of large model in Table 3. Except for HellaSwag, QuaDMix-BMK outperforms QuaDMix-OH on other tasks, which agrees with the trend on prediction loss. The inconsistent conclusion on HellaSwag is because the predict loss has larger variance as mentioned above, making the proxy ability lower than other tasks. How to further increase the proxy ability is one of the future direction to explore.

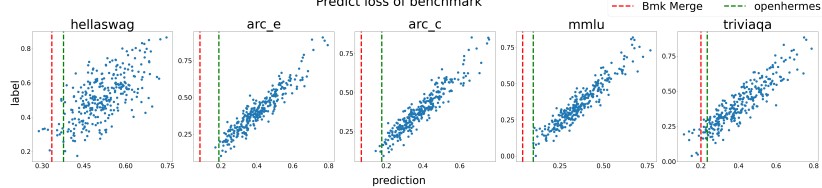

Figure 4: The prediction loss of QuaDMix-BMK surpasses QuaDMix-OH on all 5 downstream tasks.

# 7 CONCLUSION

In this paper, we propose a novel data selection method QuaDMix that jointly optimizes the data quality and diversity for language model pretraining. We design a parameterized space that controls both the data quality and diversity, and conduct proxy experiments to find the correlation between the parameter and model performance. The training data generated with optimal parameters are proved to outperform others on various downstream tasks.

## 8 ETHICS STATEMENT

Our research is based on the publicly available and extensively filtered RefinedWeb dataset. We do not foresee any direct negative societal impacts stemming from our methodology or the resulting models.

## 9 REPRODUCIBILITY STATEMENT

Our experiments are based on the open-source RefinedWeb dataset. All experimental settings, model architectures, hyperparameters, and implementation details have been thoroughly described in the main body and the appendix to ensure that other researchers can independently reproduce our results based on this information.

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

# A  CRITERIA AND DOMAIN CLASSIFIER

We use three quality criteria in QuaDMix, namely AskLLM, Fineweb-Edu, and DCLM. And one domain classifier to identify which domain each datapoint belongs to. The spearman correlation between three criteria is as illustrated in Figure 5. The three criteria have pool correlation that indacating they focus on diffrent aspects of the documents.

**AskLLM** (Sachdeva et al., 2024) is based on Llama-2-7B, with the prompt shown in Figure 6, it gives every document a score $P("yes"|prompt)$.

**Fineweb-Edu** (Penedo et al., 2024) uses the prompt shown in figure 7, and DeBERTa-v3-small to give the ducument a score. It scores a document from its educational value.

**DCLM** (Li et al., 2024) is a fastText model (Joulin et al., 2017) trained with instruction-formatted data from Openhermes 2.5(Teknium, 2023) and r/ExplainLikeImFive(ELI5) as the positive data and random data from Commen Crawl as negative data.

**Domain Classifier** (Jennings et al.) is based on DeBERTa-V3-Base with 512 token context length, it classifies documents into one of 26 domains. The 26 domains are illustrated in Figure 8.

# B  MODEL STRUCTURE & TRAINING DETAILS

We use 3000 proxy-model with 1M parameter and train the large-scale model with 530M parameter. We use transformer architecture (Vaswani et al., 2017), SwiGLU (Shazeer, 2020) as the activation function and RoPE embeddings (Su et al., 2024). We use a tokenizer with 136k vocabulary. The detailed model structure are illustrated in table 4.

We train all the model with 2048 as the max sequence length, we use a cosine decay schedular and the initial learning rate is shown in table 4, the warm up ratio is set 0.5%. We use AdamW optimizer with $\beta_1 = 0.9$, $\beta_2 = 0.95$, weight-decay= 0.1. We train 1M proxy model for 1B tokens and 530M model for 1T tokens, the performance reported for 530M model is its corresponding results at 500B token for time efficience. And we train 1.2B model for 30B tokens and 6.5B model for 150B tokens.

# C  EVALUATION

We choose 9 downstream tasks, including 3 commonsense reasoning tasks (PIQA (Bisk et al., 2019), HellaSwag (Zellers et al., 2019), OpenBookQA (Mihaylov et al., 2018)), 3 reading comprehension tasks (ARC-E/C (Clark et al., 2018), TriviaQA (Joshi et al., 2017)), 1 math problem solving task (SVAMP (Patel et al., 2021)) and 2 knowledge intensive tasks (MMLU (Hendrycks et al., 2021), NQ-open (Kwiatkowski et al., 2019; Lee et al., 2019)) as the evaluation sets. We use lm-eval-harnes to evaluate the model performance on all above benchmarks except MMLU. And we evaluate performance of MMLU using LightEval library (Habib et al., 2023).

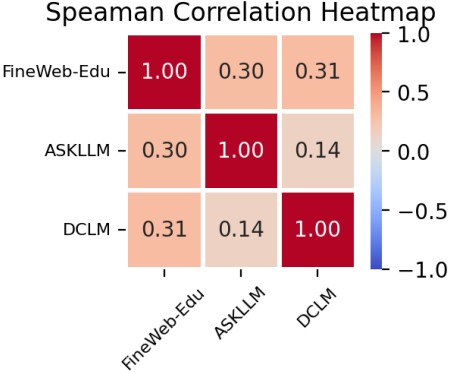

Figure 5: The spearman correlation between three criteria.

```
                              ASKLLM Prompt

###

This is a pretraining .... datapoint.

###

Does the previous paragraph demarcated within ### and ### contain informative signal for pre-
training a large-language model? An informative datapoint should be well-formatted, contain
some usable knowledge of the world, and strictly NOT have any harmful, racist, sexist, etc.
content.

OPTIONS:

- yes

- no
```

Figure 6: The Prompt of ASKLLM.

## D  COST ANALYSIS

The time cost for each model of different size is illustrated in table 6. Thought using three criteria
and one domain classifier cost extra GPU time, the results have a wide range of applications in terms
of quality analysis beyond this work, We do not include these costs in our calculation. For the 3000
proxy models, 3000 H-100 GPU hours were spent, it's about half of the cost of training a 7B model
for 150B token. Considering the improvements QuaDMix can bring, we believe it is worthwhile.

## E  DETAILED EXPERIMENT RESULTS

We show the full experiment results on 530M model in table 7, note that the performance of 530M
models reported in this paper is its corresponding results at 500B token for time efficience thought
the training scheduler was set 1T token. We also conducted experiments with other scale models to
further verify the effectiveness of QuaDMix, which is illustrated in table 8.

## F  PROXY ABILITY OF SMALL MODELS

We use Openhermes 2.5 as the evaluation set for proxy models, for efficiency, we sampled 10k sam-
ples from it, named Openhermes-10k. And train a regression model taking the QuaDMix parameters
as input and the corresponding normalized evaluation loss as output. And the 3000 experiments are
split into 2800 and 200 for training and evaluation. We tested on three regression models: Light-
GBM, SVR (Drucker et al., 1996) with Linear kernel and SVR with RBF kernel. As illustrated in

Table 4: Structure of models used in QuaDMix.

| Hyperparameter | 1M | 530M | 1.2B | 7.7B |
|---|---|---|---|---|
| Hidden states dimention | 192 | 1536 | 2048 | 4096 |
| MLP dimention | 768 | 2816 | 5540 | 14336 |
| Number of Layers | 3 | 24 | 24 | 32 |
| Number of Heads | 3 | 16 | 16 | 32 |
| Initial Learning Rate | 0.005 | 0.0008 | 0.0005 | 0.000036 |
| Batch Size | 128 | 1024 | 1024 | 4096 |

Table 5: Summary of Downstream Evaluation Tasks

| Downstream Task | Category | Metric | Shots | Baseline |
|---|---|---|---|---|
| PIQA | Commonsense Reasoning | acc_norm | 5 | 0.50 |
| HellaSwag | Commonsense Reasoning | acc_norm | 10 | 0.25 |
| OpenBookQA | Commonsense Reasoning | acc_norm | 10 | 0.25 |
| ARC-Easy | Reading Comprehension | acc_norm | 25 | 0.25 |
| ARC-Challenge | Reading Comprehension | acc_norm | 25 | 0.25 |
| TriviaQA | Reading Comprehension | exact_match | 5 | 0.00 |
| SVAMP | Math Problem Solving | exact_match | 5 | 0.00 |
| MMLU(lighteval) | Knowledge Intensive | acc_norm | 5 | 0.25 |
| NQ-open | Knowledge Intensive | exact_match | 5 | 0.00 |

## Fineweb-Edu Prompt

Below is an extract from a web page. Evaluate whether the page has a high educational value and could be useful in an educational setting for teaching from primary school to grade school levels using the additive 5-point scoring system described below. Points are accumulated based on the satisfaction of each criterion:

- Add 1 point if the extract provides some basic information relevant to educational top- ics, even if it includes some irrelevant or non-academic content like advertisements and promotional material.
- Add another point if the extract addresses certain elements pertinent to education but does not align closely with educational standards. It might mix educational content with non-educational material, offering a superficial overview of potentially useful topics, or presenting information in a disorganized manner and incoherent writing style.
- Award a third point if the extract is appropriate for educational use and introduces key concepts relevant to school curricula. It is coherent though it may not be comprehensive or could include some extraneous information. It may resemble an introductory section of a textbook or a basic tutorial that is suitable for learning but has notable limitations like treating concepts that are too complex for grade school students.
- Grant a fourth point if the extract highly relevant and beneficial for educational purposes for a level not higher than grade school, exhibiting a clear and consistent writing style. It could be similar to a chapter from a textbook or a tutorial, offering substantial educational content, including exercises and solutions, with minimal irrelevant information, and the concepts aren't too advanced for grade school students. The content is coherent, focused, and valuable for structured learning.
- Bestow a fifth point if the extract is outstanding in its educational value, perfectly suited for teaching either at primary school or grade school. It follows detailed reasoning, the writing style is easy to follow and offers profound and thorough insights into the subject matter, devoid of any non-educational or complex content.

The extract: <EXAMPLE>.

After examining the extract:

- Briefly justify your total score, up to 100 words.

- Conclude with the score using the format: "Educational score: <total points>"

Figure 7: The Prompt of Fineweb-Edu.

figure 9, the left figure indicates that as the training data increases, the Mean Average Error (MAE) of the regression models on the validation set continues to decrease, and it can be observed that the performance of LightGBM is consistently the best. The right figure shows strong correlation between the predicted loss and the real model loss on the validation set, providing the evidence that there exists statistical pattern between the QuaDMix parameters and the model performance.

We test the evaluation loss of proxy models on various dnowstream tasks, their pearson correlations between loss and predicted loss of lightGBM on validation set are illustrated in table 9. There is a

## Domains

- Adult
- Arts_and_Entertainment
- Autos_and_Vehicles
- Beauty_and_Fitness
- Books_and_Literature
- Business_and_Industrial
- Computers_and_Electronics
- Finance
- Food_and_Drink
- Games
- Health
- Hobbies_and_Leisure
- Home_and_Garden
- Internet_and_Telecom
- Jobs_and_Education
- Law_and_Government
- News
- Online_Communities
- People_and_Society
- Pets_and_Animals
- Real_Estate
- Science
- Sensitive_Subjects
- Shopping
- Sports
- Travel_and_Transportation

Figure 8: The 26 domains of Domain Classifier.

Table 6: Training cost for different model configurations.

| Model Size | Token | H-100 GPU Hours |
|------------|-------|-----------------|
| 1M | 1B | $\sim$1 |
| 530M | 500B | $\sim$2300 |
| 1.2B | 30B | $\sim$220 |
| 7.7B | 150B | $\sim$5900 |

significant spearmanr correlation between ground truth and the predicted loss by lightGBM on the validation set for vast majority of benchmarks, except for a few benchmarks such as HellaSwag or Xnli. We believe this is because some factors other than the QuaDMix parameters can also affect performance on these benchmarks. And this will be left for our future work.

## G TARGETED OPTIMIZATION

To further verify QuaDMix's ability to optimize for specific dnownstream tasks, we test the evaluation loss of proxy models on various dnownstream tasks. For each benchmark, we have 3000 evaluation loss from proxy models, we treat it as a 3000-dimentional vector thus we can calculate the correlation between benchmarks. We show the spearmanr correlation between all the benchmarks in figure 10.

**Benchmark Merge** We choose 5 benchmarks: HellaSwag, ARC-E, ARC-C, MMLU, TriviaQA and train a lightGBM for each benchmark. We aim to find a set of QuaDMix parameters so that the pretrain model perform as well as possible on these 5 benchmarks simultaneously. Specifically, we find a set of weights $\alpha$ for different benchmarks and a set of QuaDMix parameters so that the $\alpha$ weighted sum of the losses predicted by LightGBM under the QuaDMix parameters for the 5 benchmarks is minimized. As illustrated in table 7, the QuaDMix-BMK outperforms QuaDMix-OH on almost all above 5 benchmarks except HellaSwag, the QuaDMix-BMK's lagging behind on HellaSwag

Table 7: Performance comparison of different methods across various benchmarks of 530M model.

| Method | PIQA | HellaSwag | OpenBookQA | ARC-E | ARC-C | Triviaqa | SVAMP | MMLU | NQ-open |
|--------|------|-----------|------------|-------|-------|----------|-------|------|---------|
| **Random Selection** | 70.9 | 51.1 | 32.7 | 57.2 | 27.9 | 13.6 | 2.8 | 29.4 | 5.4 |
| **DSIR** | 70.9 | 45.9 | 30.8 | 60.9 | 30.2 | 13.6 | 6.9 | 30.1 | 4.8 |
| **RegMix** | 71.4 | 52.1 | 33.6 | 62.6 | 31.1 | 12.7 | 3.5 | 30.1 | 5.3 |
| **Fineweb-edu** | 73.7 | 54.8 | 34.4 | 70.8 | 38.2 | 18.1 | 10.7 | 34.2 | 6.9 |
| **ASKLLM** | 73.2 | 54.2 | 35.1 | 64.8 | 32.2 | 19.7 | 2.3 | 30.9 | 7.1 |
| **DCLM** | 71.2 | 51.7 | 36.3 | 68.5 | 35.7 | 19.4 | 8.2 | 32.4 | 7.2 |
| **Criteria Mix** | 72.6 | 54.0 | 34.5 | 67.2 | 34.2 | 18.9 | 3.1 | 32.5 | 7.4 |
| **QuaDMix-OH** | 73.2 | **56.5** | **37.3** | 71.1 | 39.2 | 21.6 | 10.2 | 34.1 | 7.9 |
| **QuaDMix-BMK** | **74.1** | 56.1 | 36.9 | **71.3** | **40.2** | **22.8** | **11.5** | **34.4** | **8.1** |

Table 8: Performance comparison of different model size across various benchmarks.

| Method | Model-size | PIQA | HellaSwag | OpenBookQA | ARC-E | ARC-C | Triviaqa | SVAMP | MMLU | NQ-open | Avg |
|---|---|---|---|---|---|---|---|---|---|---|---|
| Random Selection | 1.2B | 71.82 | 50.93 | 31.40 | 57.70 | 27.90 | 12.61 | 3.90 | 29.73 | 5.29 | 32.36 |
| QuaDMix-OH | 1.2B | 74.05 | 57.20 | 38.00 | 71.09 | 39.76 | 21.43 | 7.80 | 34.75 | 7.73 | 39.09 |
| Random Selection | 7B | 73.18 | 53.69 | 31.80 | 59.68 | 29.27 | 16.13 | 1.60 | 30.60 | 5.54 | 33.50 |
| QuaDMix-OH | 7B | 75.73 | 59.14 | 38.80 | 71.17 | 40.61 | 24.41 | 5.60 | 35.62 | 8.39 | 39.94 |

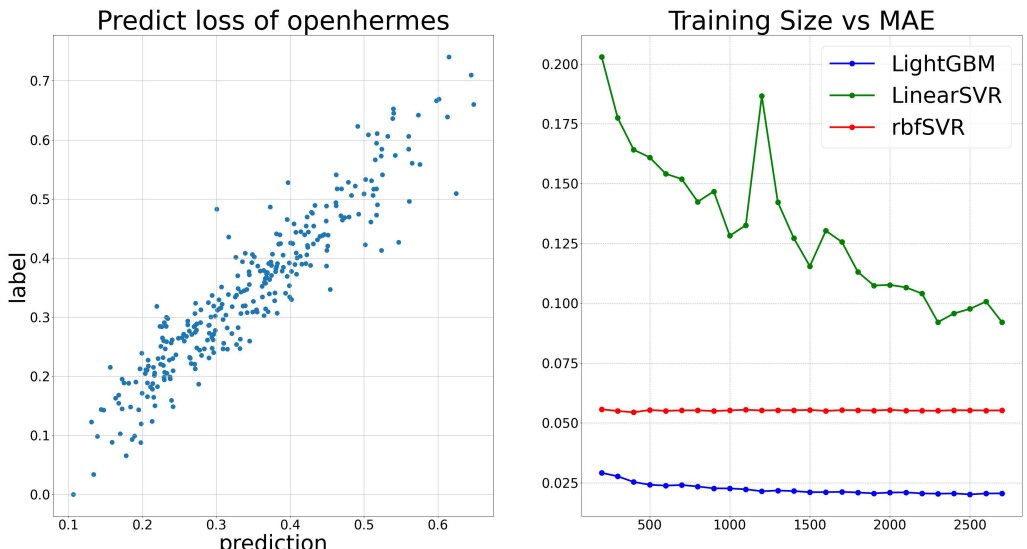

Figure 9: Left: The prediction model loss vs real model loss. Right: The regression model performance (MAE) vs training size.

Table 9: Pearson correlation across different benchmarks.

| Benchmark | Pearson | Benchmark | Pearson | Benchmark | Pearson |
|---|---|---|---|---|---|
| ARC-C | 0.9451 | XCOPA-EN | 0.8972 | PIQA | 0.9460 |
| ARC-E | 0.9430 | Social-I-QA | 0.9040 | MathQA | 0.8965 |
| MMLU | 0.9452 | MuSR | 0.9036 | OpenHermes | 0.9545 |
| TriviaQA | 0.9251 | CommonsenseQA | 0.8882 | GSM8K | 0.7055 |
| BoolQ | 0.9128 | XWinograd | 0.8732 | Minerva (Algebra) | 0.9549 |
| NQ-Open | 0.8654 | XNLI-EN | 0.7514 | APE210K | 0.8921 |
| XStoryCloze | 0.9014 | HellaSwag | 0.6605 | OpenBookQA | 0.9045 |

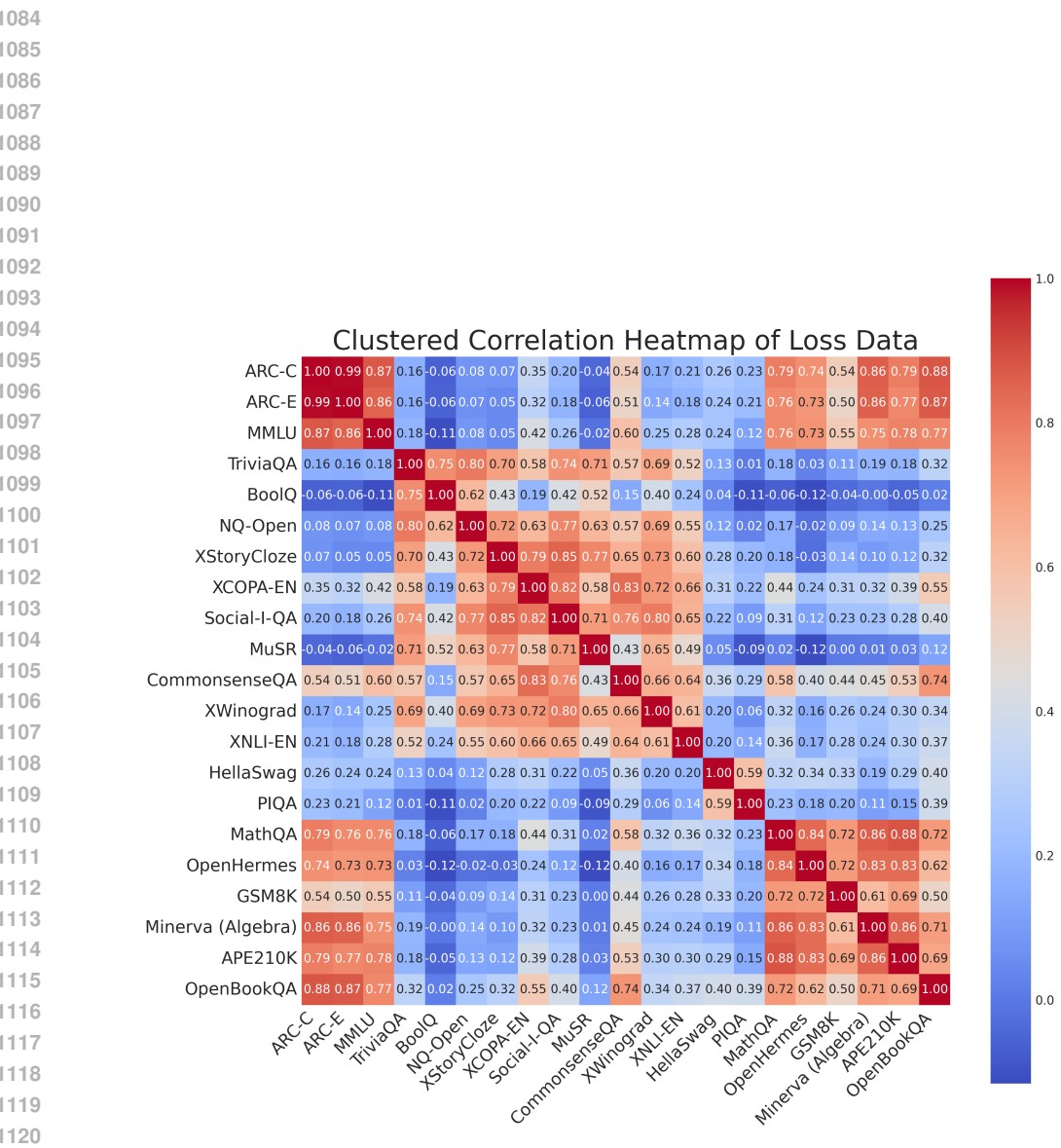

Figure 10: The spearmanr correlation between benchmarks.

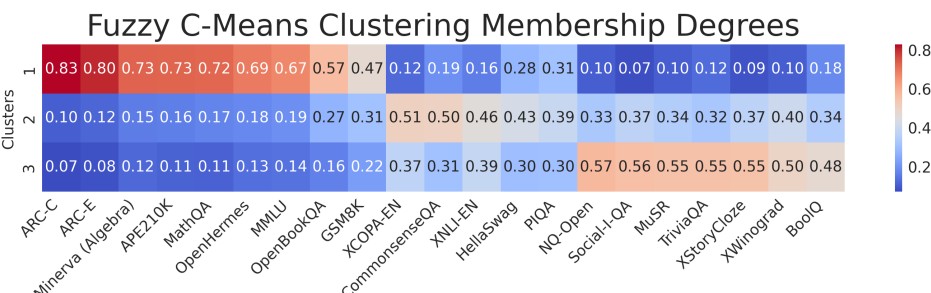

Figure 11: The spearmanr correlation between enchmarks.

may be due to the LightGBM of this benchmark having poorer predictive ability compared to other benchmarks, see Appendix F.

**Cluster Merge** As shown in figure 10, there is a clear clustering phenomenon among different benchmarks, where the same set of QuaDMix parameters model performs similarly on benchmarks within the same cluster. We believe this is because benchmarks in the same cluster require similar abilities, such as mathematical ability, logical reasoning ability, and reading comprehension ability. Therefore, we further optimize the Benchmark Merge method to enable QuaDMix to simultaneously optimize multiple abilities.

We first use fuzzy C-means to cluster all benchmarks into N clusters. Fuzzy c-maens is a soft clustring method that allows each datapoint belongs to all cluster and represents the "fuzziness" of each data point's membership in each cluster by assigning a membership degree between 0 and 1. As illustrated in figure 11, we show the cluster results of $N = 3$ in 21 benchmarks. Then we normaliz the weights in a cluster, so we can get a weighted score for each cluster with a set of QuaDMix parameters. And we finally find a optimal weights of cluter and a set of QuaDMix parameters that minimize the weighted cluter loss.

We also observed that when using Openhermes as the only validation set, the mathematics related benchmarks' performance have also improved like MathQA, GSM8K, etc. They are all in the same cluster from 11, indicating that different benchmarks may reqiure similar abilities, and Cluster Merge gives it a way to targeted optimize the model performance based on the abilities required by the benchmarks.

We show the lightGBM predicte loss for each method in figure 12, when using 5 benchmarks (HellaSwag, ARC-E, ARC-C,MMLU, TriviaQA) as evaluation set with Benchmark Merge, it has a great improvement in target benchmarks. As for the cluster merge method, we obersed a highly competitive result across all benchmarks, validating the feasiblity of our Cluster Meger for optimize model performance based on abilities rather than benchmarks.

# H    LIMITATIONS

We note several limitations of our work. There exist improvement space for the design of parameter space of QuaDMix. For example the parameters of sampling function may generate similar functions under different parameters, which will cause redundancy and introduce uncertainty into the regression model. Secondly, the searching in the parameter space for optimal parameters is inefficient. We use random guessing in a space with 200 more dimensions, for certain the current optimal parameter is a local minimum and how to effectively search in the parameter space remains unclear. Finally, the proxy ability of small models is crucial, what is the systematic way to improve it is an important yet less explored topic. However, QuaDMix provides a useful solution for jointly optimize for data quality and diversity, and it worth continually exploring on the limitations mentioned above.

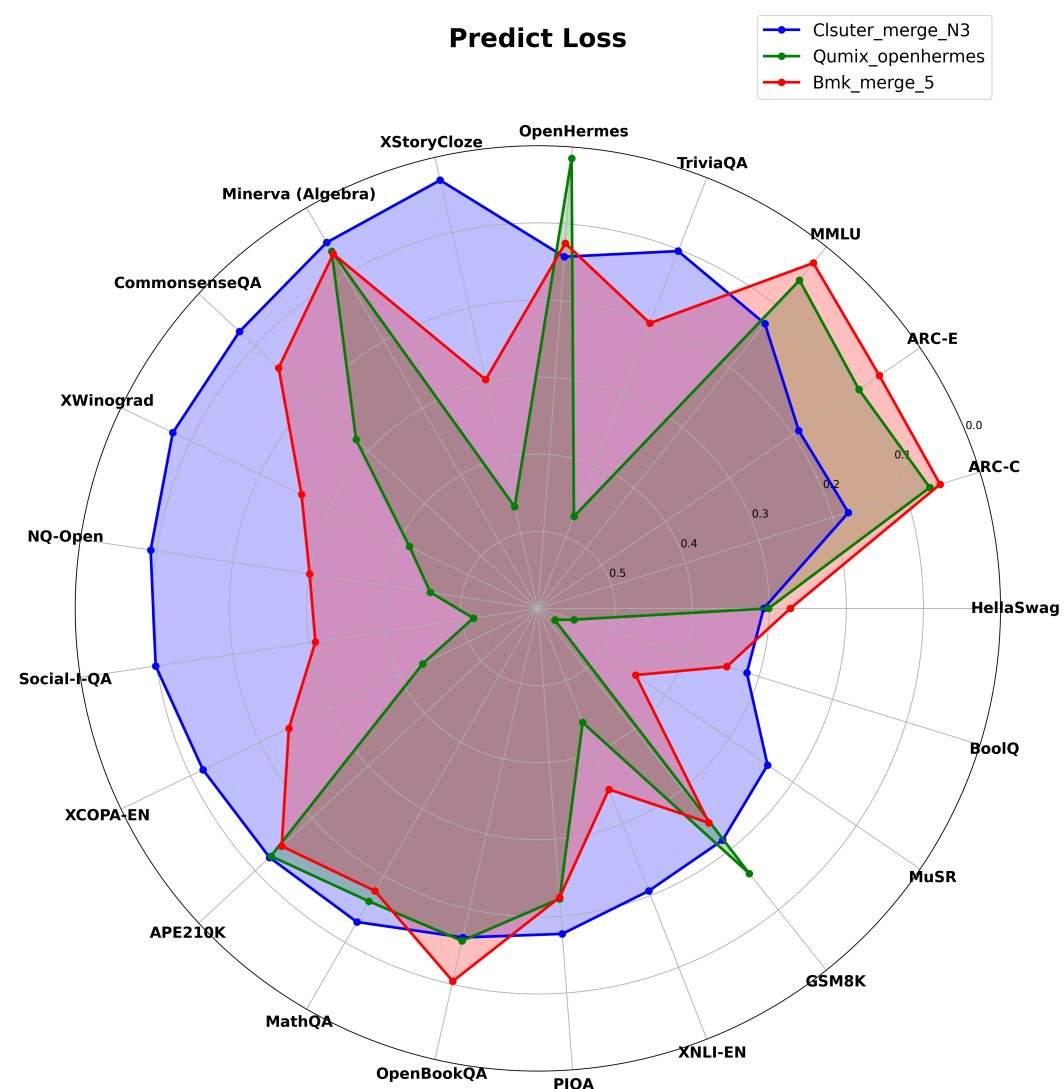

Figure 12: The spearman correlation between benchmarks.

# I    USAGE OF LLM

During the writing process of this paper, large language models (LLMs) were used for language polishing and spell-checking. All academic content is the original work of the authors.

