# OpenReview forum: "QUADMIX: QUALITY-DIVERSITY BALANCED DATA SELECTION FOR EFFICIENT LLM PRETRAINING"
_ICLR.cc/2026/Conference — ICLR 2026 Conference Withdrawn Submission_

### Official Review · Reviewer_oVFF · 2025-10-21

**Soundness:** 2
**Presentation:** 2
**Contribution:** 2
**Rating:** 4
**Confidence:** 4

**Summary:**

This paper introduces QuaDMix, a unified data selection framework that jointly optimizes data quality and diversity for efficient LLM pretraining. The authors identify an important challenge: existing approaches typically optimize quality and diversity separately, overlooking their inherent trade-offs when working with fixed training budgets. The proposed method achieves 7.2% average improvement over random selection at 500B tokens.

**Strengths:**

- The research topic of this paper is timely and interesting.
- The paper provides clear exposition of the methodology with helpful visualizations, particularly Figure 2 which effectively illustrates the three-stage pipeline.
- The experimental methodology is comprehensive, involving 3,000 proxy model experiments, multiple model scales (1M to 7.7B parameters), and evaluation across 9 diverse downstream tasks.

**Weaknesses:**

1. While the paper claims efficiency, the full preprocessing costs are not transparently reported. Training 3,000 proxy models requires 3,000 H-100 GPU hours, and applying three quality scorers plus domain classification to the entire dataset incurs additional substantial costs that are dismissed as "having applications beyond this work." A fair comparison should include all costs from raw data to final model.

2. The entire framework depends critically on a specific 26-domain classifier. There is no investigation about:
- sensitivity to domain granularity (what about 10 or 50 domains?).
- robustness to different domain taxonomies.
- what happens when domains are ill-defined or overlapping.

3. The sigmoid-based sampling function (Equation 3) appears unmotivated. Why this specific form? How sensitive are results to alternative functional forms (e.g., exponential decay, piecewise linear)?

4. Critical ablations are missing:
- Uniform vs. learned weights for quality criteria merging
- Domain-specific vs. global parameters for sampling
- Contribution of individual components (quality merging, domain-specific parameters, sampling function design)

5. The paper acknowledges HellaSwag has poor proxy ability (Section 6, Figure 4) but doesn't investigate root causes or solutions. This raises concerns about which other benchmarks might have similar issues that haven't been discovered.

6. Table 7 reports 530M model results at 500B tokens though "the training scheduler was set 1T token" (Appendix E). This inconsistency makes comparisons less rigorous and raises questions about whether models were fully converged.

7.  A lack of more recent works as related works or comparison baseline methods:

- [1] Harnessing Diversity for Important Data Selection in Pretraining Large Language Models  ICLR 2025
- [2] MASS: Mathematical Data Selection via Skill Graphs for Pretraining Large Language Models https://arxiv.org/abs/2503.14917
- [3] Meta-rater: A Multi-dimensional Data Selection Method for Pre-training Language Models ACL 2025

Since [1] also focuses on diversity, this paper should compare with [1].

**Questions:**

- What is QuaDMix's performance on other pretraining corpora (C4, The Pile, RedPajama-v2)? Do optimal parameters transfer across datasets or must they be recomputed for each?

- How do results change with different domain classifiers or domain granularities?

- Have you investigated why HellaSwag has poor proxy ability? Is it related to the task format, the evaluation metric, or something else? Are there other benchmarks with similar issues that haven't been identified?

---

### Official Review · Reviewer_73zH · 2025-10-27

**Soundness:** 3
**Presentation:** 2
**Contribution:** 3
**Rating:** 4
**Confidence:** 3

**Summary:**

This paper introduces QuaDMix, a unified framework that jointly optimizes data quality and diversity for efficient LLM pretraining. Unlike existing approaches that treat quality filtering and diversity optimization separately, QuaDMix addresses their inherent trade-off through a parameterized sampling function that determines each data point's sampling probability based on multiple quality criteria and domain classifications. The framework employs domain-specific parameters to merge quality scores from different filters and control sampling frequency, using a sigmoid-based function that prioritizes higher-quality data while maintaining domain diversity. To efficiently search the high-dimensional parameter space, the authors train thousands of small proxy models and use LightGBM regression to predict optimal parameters, avoiding costly large-scale experiments. Extensive experiments demonstrate that QuaDMix achieves an average 7.2% performance improvement across multiple benchmarks while being 27 times more efficient than random selection, reaching equivalent performance with only 18B tokens versus 500B tokens. The method also enables task-specific optimization by adjusting evaluation targets and reveals that different quality criteria exhibit complementary strengths, with merged criteria consistently outperforming individual filters.

**Strengths:**

Originality: The paper presents a novel and systematic framework, QuaDMix, that jointly optimizes data quality and diversity for LLM pretraining—addressing a critical and underexplored trade-off in data selection. The proposed parameterized sampling function and multi-criteria quality integration offer a fresh perspective beyond conventional sequential filtering approaches.
Quality: The work is supported by comprehensive experimental validation, including large-scale proxy model training and rigorous evaluation across multiple model sizes and benchmarks. The consistent performance gains and thorough ablation studies strongly demonstrate the method’s robustness and reliability.
Clarity: The paper is well-structured and accessible. The system architecture is clearly illustrated through a dedicated workflow diagram , and the sampling mechanism is precisely formulated with supporting equations, enabling straightforward interpretation and reproducibility.
Significance: By explicitly modeling the interplay between data quality and diversity, QuaDMix enables more representative and efficient data curation for LLM training. The framework’s ability to improve performance across diverse downstream tasks underscores its practical value and potential impact on future pretraining paradigms.

**Weaknesses:**

1.Lack of systematic ablation on key parameters.
While the paper defines a parameterized sampling function with multiple parameters (λ, ω, η, ϵ), the ablation studies only explore ω and the quality merging weights α. Other crucial parameters, such as λ, which controls the steepness of the sampling curve, are not analyzed. Without such experiments, it is difficult to understand how sensitive the model’s performance is to these parameters or how they influence the balance between quality and diversity.
2.Inconsistent figure and table referencing.
In Section 5.1 (Training and Evaluation), the paper mentions experiments conducted with a 7B-parameter model, but the corresponding results are not presented in the main text and appear only in the appendix. This inconsistency in figure and table placement weakens the clarity and completeness of the experimental presentation. The authors should ensure that all key results are either summarized in the main body or properly referenced to the appendix.
3.Lack of validation on different datasets.
All experiments are conducted exclusively on the RefinedWeb dataset. While this provides a clean evaluation environment, it leaves open the question of whether QuaDMix generalizes well to other large-scale pretraining corpora. Evaluating the proposed framework on diverse datasets would significantly strengthen the paper’s claims of generality and robustness.

**Questions:**

1.Why were certain parameters, such as λ, not included in the ablation studies? Was this omission due to computational resource constraints, or based on prior assumptions about their impact?
2.Could the references to figures and tables be made more comprehensive and precise to improve clarity?
3.Could the reported conclusions be potentially influenced by dataset bias? In particular, is the performance of QuaDMix sensitive to the specific characteristics of the RefinedWeb dataset?

---

### Official Review · Reviewer_6Q5c · 2025-11-01

**Soundness:** 2
**Presentation:** 3
**Contribution:** 2
**Rating:** 4
**Confidence:** 4

**Summary:**

This paper introduces QuaDMix, a unified data selection framework for LLM pretraining that jointly optimizes data quality and diversity. The key idea is using a parameterized sampling function that integrates multiple quality criteria and domain classification to determine sampling probabilities for each data point. The authors train 3000 proxy models (1M parameters) with different parameter configurations, then use LightGBM regression to predict optimal parameters for large-scale training. Experiments on 530M, 1.2B, and 7B parameter models demonstrate that QuaDMix achieves 7.2% average performance improvement over baselines that optimize quality or diversity independently. The framework also supports task-specific optimization by selecting appropriate validation sets during proxy training.

**Strengths:**

1. Problem Formulation: The paper clearly articulates the trade-off between data quality and diversity, and makes a compelling case for joint optimization. The observation that different quality criteria bias domain distributions (Figure 1b) effectively motivates the unified approach.
2. Experimental Validation: Extensive experiments across multiple scales (530M, 1.2B, 7.7B parameters)
Evaluation on 9 diverse downstream tasks spanning commonsense reasoning, reading comprehension, math, and knowledge-intensive benchmarks. Strong ablations demonstrating the value of merging quality criteria (Table 2) and the importance of the quality-diversity trade-off
3. Practical Impact: The 27× training efficiency improvement (achieving baseline performance at 18B vs 500B tokens) represents significant computational savings. The framework is also flexible enough to optimize for specific downstream tasks (QuaDMix-BMK).

4. Reproducibility: The paper provides comprehensive details that will facilitate the reproduction and extension of the work. Appendices detail model architectures, training hyperparameters, cost analysis (Table 6), and implementation specifics necessary for replication. The analysis of proxy model correlation across 21 benchmarks (Table 9, Appendix F) and targeted optimization procedures (Appendix G) offer meaningful insights for practitioners adapting the work. The methodological contribution of extending scaling laws to data composition—demonstrating how regression models can predict large-model performance from small-scale proxy experiments is clearly articulated with sufficient detail (Algorithm 1, parameter sampling procedures) to enable future researchers to apply and build upon this approach in their own data selection pipelines.

**Weaknesses:**

1. Insufficient methodological justification: quality percentile estimation using only 10,000 documents (Section 3.4) may introduce noise without error analysis; Algorithm 1's parameter rescaling factors (10³, 0.1, 1/1000) appear arbitrary and unjustified; sigmoid-based sampling function (Equation 3) is one choice among many without ablation on alternative functional forms; parameter space has acknowledged redundancies where different settings yield similar distributions, complicating regression learning.
2. Presentation and clarity issues: Figure 2 unclear about information flow and where optimization occurs; Table 1 compares methods with vastly different token budgets hindering interpretation; "top 5%" quality threshold finding needs deeper investigation of consistency across domains; notation inconsistencies (θ_m vs. θ); evaluation metrics not explicitly defined; baseline acronyms not always clear; task groupings could be more explicit.
3. Lack of statistical tests: no error bars or confidence intervals on any results; single training run per configuration without variance analysis from random initialization; Pearson correlation used but Spearman may be more appropriate for ranking metrics; no significance testing between methods; 95.45% correlation claimed but confidence bounds not provided.
4. Limited scope and analysis depth: only evaluated on pretraining-from-scratch, not fine-tuning or instruction-tuning stages; no analysis of data selection effects beyond accuracy (calibration, robustness, fairness, biases); missing discussion of how optimal domain proportions might vary at different training scales; no investigation of learned domain mixtures vs. baselines to understand what "optimal diversity" actually means; cost-benefit analysis incomplete (total cost including quality scoring not provided).

**Questions:**

1. Have you considered more efficient hyperparameter optimization methods? What is the sensitivity of final performance to the number of proxy experiments?
2. Can you provide analysis on why certain benchmarks (HellaSwag) have poor proxy correlation? Are there task characteristics that predict proxy reliability?
3. How sensitive are results to the choice of domain taxonomy? What happens with 10 vs. 50 domains?
4. Would the framework work with different quality metrics (e.g., perplexity-based, model-based scoring)? Is there a principled way to select which quality criteria to include?
5. The parameter space has clear redundancies (as acknowledged). Have you explored dimension reduction techniques or parameter tying?
Can you provide the actual domain mixture learned by QuaDMix vs. baselines to better understand what "optimal diversity" looks like?

---

### Official Review · Reviewer_ufJ1 · 2025-11-01

**Soundness:** 3
**Presentation:** 3
**Contribution:** 3
**Rating:** 4
**Confidence:** 3

**Summary:**

This paper addresses the trade-off between data quality and diversity in LLM pretraining, a gap in existing work that optimizes the two metrics separately. It proposes QuaDMix, a unified data selection framework: first, it uses 3 quality scorers (AskLLM, Fineweb-Edu, DCLM) and a domain classifier (26 domains) to label data; then, it designs a parameterized sampling function (merging quality scores via domain-specific weights and adjusting sampling frequency with sigmoid-based logic) to balance quality and diversity. To find optimal parameters efficiently, it trains 3,000 small proxy models (1M parameters) on sampled datasets, uses LightGBM to fit parameter-performance correlations, and predicts optimal parameters. Experiments on RefinedWeb show QuaDMix outperforms baselines (e.g., Random Selection, Fineweb-edu) by 7.2% on average across 9 downstream tasks, with 27x faster convergence.

**Strengths:**

1. QuaDMix unifies quality-diversity optimization for LLM pretraining via parameterized sampling.
2. It uses proxy models + LightGBM for efficient parameter search, cutting costs.
3. QuaDMix outperforms baselines across models/tasks and enables task optimization.

**Weaknesses:**

1. QuaDMix only tests on RefinedWeb, lacking cross-dataset validation to prove adaptability to other LLM pretraining data.
2. The 26-domain classifier lacks justification for optimality, with no validation on alternative diversity categorizations.

**Questions:**

As shown in weakness.

---

### Note · Authors · 2026-01-26

I have read and agree with the venue's withdrawal policy on behalf of myself and my co-authors.

---

### Meta-Review · Area_Chair_QPvA · 2026-01-07

**Summary:**

Reviewers raised concerns about the conceptual novelty and positioning of the proposed quality-diversity balanced data selection strategy, noting significant overlap with prior work on data filtering, curriculum learning, and mixture-based data selection for LLM pretraining. While the idea is intuitively appealing, reviewers questioned whether QUADMIX introduces fundamentally new principles or algorithms beyond a particular instantiation of existing heuristics. Additional concerns focused on the limited theoretical justification and whether the reported empirical gains generalize across different model scales, data regimes, and training objectives.

**Reviewer Concerns:**

The rebuttal clarified the design choices of QUADMIX and provided additional discussion on how quality and diversity signals are combined in practice, addressing some questions about implementation and empirical setup. These responses helped improve clarity and reduced ambiguity around experimental comparisons. However, the central concerns regarding the incremental nature of the contribution, lack of strong theoretical grounding, and insufficient evidence of broad generality remain largely unresolved. In particular, the rebuttal did not convincingly demonstrate that the proposed approach yields insights or benefits that go substantially beyond existing data selection methods.

**Reviewer Scores:**

4444. Had reviewers been able to fully participate in the post-rebuttal discussion, some might have modestly increased their scores to reflect improved clarity and responsiveness. Nevertheless, it is unlikely that such changes would have altered the overall recommendation, as the main concerns about novelty and impact would likely have persisted.

---

### Decision · Program_Chairs · 2026-01-26

Reject